# Beyond Screen Time: The Different Longitudinal Relations between Adolescents’ Smartphone Use Content and Their Mental Health

**DOI:** 10.3390/children10050770

**Published:** 2023-04-24

**Authors:** Shunsen Huang, Xiaoxiong Lai, Yajun Li, Yang Cui, Yun Wang

**Affiliations:** 1State Key Laboratory of Cognitive Neuroscience and Learning, Faculty of Psychology, Beijing Normal University, Beijing 100875, China; 2Guangming Institute of Education Sciences, Shenzhen 518107, China

**Keywords:** smartphone use content, adolescent mental health, specification curve analysis, depression, anxiety, somatization

## Abstract

**Purpose**: Previous studies focusing on the relationship between adolescents’ screen time and mental health have uncovered contradictory results. By focusing on smartphone use content (SUC), this study uses specification curve analysis to explore the different effects of SUCs on mental health-based on longitudinal data. **Methods**: A total of 2552 adolescents were surveyed in the first (July 2020) and second year (April 2021). A total of 2049 eligible participants (average age = 14.39 ± 2.27, female = 1062) are included in the analysis. Participants reported 20 types of content used by them during smartphone use and their mental health (depression, anxiety, and somatization). Specification curve analysis was used to examine the longitudinal relationship between SUCs and their mental health. **Results**: Smartphone use for listening to music (median β = 0.18, *p* < 0.001, NSRPD = 25/27, *p* < 0.05), chatting online (median β = 0.15, *p* < 0.001, NSRPD = 24/27, *p* < 0.05), watching TV (median β = 0.14, *p* < 0.001, NSRPD = 24/27, *p* < 0.05), and playing games (median β = 0.09, *p* < 0.001, NSRPD = 19/27, *p* < 0.05) produce high to medium negative effects on subsequent mental health. Only using smartphones for online courses exerts no effect on their subsequent mental health (median β = 0.01, *p* > 0.05, NSRPD = 0/27, *p* > 0.05). The left 15 types of smartphone content showed unstable effects on future mental health. Depending on the types of content used, these effects ranged from high, medium, and small to none. The relatively descending order of effect on mental health is listening to music, chatting online, watching TV, playing games, and types of content (e.g., browsing social media, making payments, reading online novels) with high but unstable effects, types of content with medium (e.g., browsing news and posting/sharing) but unstable effects, types of content (e.g., using the camera, obtaining life information, and making calls) with small but unstable effects, such as finishing homework and taking online courses. **Conclusions**: This study enlightens researchers and policymakers to update their understanding of adolescents’ technology use, especially to adopt a differentiated attitude towards different media use content. As nutritionists often do, a “nutritionally balanced” digital diet for young people should be recommended to the public, rather than just suggesting limits on the amount of time they can spend using digital media.

## 1. Introduction

From 2016 to 2022, global smartphone use has increased, with nearly 6.57 billion people worldwide becoming smartphone users in 2022 [1]. According to the 46th Statistical Report on Internet Development in China, as of April 2020, approximately 169 million Chinese individuals under 19 years old access the Internet via smartphones [2]. Through smartphones, people can play online games, chat with friends, watch videos, access social media, search for information, shop online, and more. These functions facilitate people’s daily lives. However, academic researchers have debated whether digital technology improves adolescents’ well being or has a negative or no influence on their mental health.

The extant research on the topic of the effect of digital technology on adolescents’ mental health or well being has uncovered controversial results. Some researchers have found that adolescents’ media use may promote their mental health, as Valkenburg et al. found that receiving positive feedback on social networking sites may improve adolescents’ mental health [3]. Other researchers have argued that digital media use may exert a weak effect on adolescents’ mental health [4]. Specification curve analysis (SCA) has been recently used to explore the relationship between screen time and adolescents’ well being and found little evidence of substantial negative associations, they believe that these effects are too small to warrant policy change [5]. A host of researchers have argued that adolescents’ digital media use harms their well being [6,7]. Several recent longitudinal studies have shown that screen time [8], the frequency of Facebook use [9], social media use time [10], and internet use time [11] negatively predicted adolescents’ future well being. Even so, many of the existing studies on the relationship between smartphone/social media use and mental health were cross-sectional and are vulnerable to reverse causation issues [12].

There exist some gaps in this field that may contribute to these controversial debates. First, screen time is no longer a useful construct, but it still dominates research and public discourse, and, thus, changing the conversation to more accurately reflective aspects is necessary [13], as researchers have argued that it is more likely that what adolescents post and view online rather than screen time (time children spend on digital media) is associated with low mental health [14]. For example, more nuanced studies of adolescents’ online activities have shown that it is not the frequency but the type of social media use that is associated with their depression [15]. Second, to some extent, these controversial results may reflect a crisis of reproducibility in psychological science due to selective reporting, selective analysis, and inadequate description of the conditions necessary to obtain the results [16]. For example, using the same datasets and the same method but different choices in selecting independent or dependent variables, researchers found a greater effect than in previous studies with the same datasets and method [17]. Specification curve analysis (SCA) can be used to alleviate this problem by exhausting all the potential combinations between variables to obtain the robustness of interesting relationships between variables [18]. Third, previous research does not distinguish between the effects of different types of digital technologies (e.g., smartphones, TVs, computers, and tablets) on adolescents’ mental health [5]. Different digital technologies with rapidly changing content may have different effects on adolescents’ mental health. As Twenge argued, some studies were conducted before smartphones became common and before the significant increase in the usage of digital media [6]. In addition, the National Study on Internet Use by Minors pointed out that 93.9% of teenagers in China access the Internet through smartphones, while the proportions of teenagers accessing the Internet through computers, laptops, TVs, and tablets are 45%, 31.5%, 56.7%, and 28.9%, respectively [19]. Therefore, it is necessary to focus on smartphones, the most influential technology in Chinese daily life. Finally, previous research using SCA based on cross-sectional data could not infer causality. No studies have applied SCA to longitudinal design in this field. Thus, a longitudinal cohort design is urged to draw valid inferences about the magnitude of different SUC effects on adolescents’ mental health.

This study uses specification curve analysis to explore the longitudinal effects of different SUCs on teens’ mental health. The complementary–interference model [20,21] argues that smartphone use has many benefits like online communication or entertainment, while using smartphones for accessing valuable information, entertainment, and communication can not only distract people from giving full attention to friends, family, and other activities in their social environment but also supplant casual (face-to-face) social interactions. Both distraction and supplantation will reduce engagement with potential sources of well being concurrently available in the environment, offset the potential benefits of smartphones, and decreases people’s well being [20,21]. Therefore, we assumed that using smartphones for communication (e.g., chatting online), accessing valuable information (e.g., using search engines), and entertainment (e.g., playing games or watching TVs) have negative effects on future mental health. Except, adolescents can also use smartphones for study purposes. The online participation theory of online learning suggests that online learning is a process of online participation that includes taking part and maintaining relations with others and collaborating with others [22]. Such participation is usually related to good academic results [22]. Thus, we hypothesize that using the smartphone for online learning purposes (e.g., using the smartphone for homework or learning) may be less likely to relate to low mental health. Therefore, the 20 types of SUC should exert different effects on adolescents’ mental health.

## 2. Materials and Methods

### 2.1. Participants

#### 2.1.1. Study Design

The present study is a two-wave cohort study, aiming at exploring the longitudinal effects of different SUCs on adolescents’ mental health. Adolescents were from 55 classes (4 or 5 random classes per school) of 12 randomly selected high schools in Henan province, China. With the help of teachers, adolescents with experience using smartphones were invited. The specific recruitment procedure is that, in the first year (July 2020) and the second year (April 2021), 2552 adolescents were surveyed, and adolescents and parents were matched by the given identification numbers and names. Nearly 95% of the adolescents in selected classes gave their responses. 

#### 2.1.2. Recruitment

In the first year, influenced by the COVID-19 pandemic, a link to the online questionnaire was sent to the participants and their parents through WeChat, a popular social media platform that is easily accessible to adolescents in China. One parent (mother or father) of each participant reported some demographic information, such as parents’ education, annual family revenue, and the subjective socioeconomic status (SES) of the family. In the second year, adolescents completed a questionnaire independently at school.

A total of 2552 parent–adolescent dyads were surveyed, and 149 dyads were excluded due to incomplete questionnaires (e.g., only providing basic demographic information (e.g., gender) in adolescents’ questionnaire) (final eligible *N* = 2403). Then, 2403 parent-adolescent dyads were surveyed in the second year, and 354 dyads with incomplete questionnaires were excluded. Therefore, the final participants contained in the later analysis totaled 2049 (average age = 14.39 ± 2.27 years, female = 1062). Little’s test (MCAR test) showed that the missing pattern (χ^2^ = 133.815, *df* = 147, *p* > 0.05) belongs to MCAR, and the average missing rate across variables was 0.12%. Thus, the EM algorithm imputation method [23] was used to process the missing data. See Table 1 for demographic information about the participants. 

#### 2.1.3. Ethical Issues

This study was approved by the IRB of the State Key Laboratory of Cognitive Neuroscience and Learning, Beijing Normal University, Beijing, China (CNL_A_0003_003; 25 July 2018). All adolescents and their parents gave their consent. 

### 2.2. Measurements

#### 2.2.1. Demographic Information 

The demographic information included age, gender, residence, only child, education level of parents (both mother and father), as well as self-reported social economic status (SES).

#### 2.2.2. Smartphone Use Content

The SUC was measured by a revised Mobile Phone Use Pattern Scale for Chinese students [24,25]. The revised questionnaire mainly measures smartphone use content related to playing games on smartphones, chatting online via smartphones, accessing news, life-related information via smartphones, etc. This revised questionnaire included 20 items (α = 0.93), and each item measured the different smartphone use content (e.g., reading online novels, playing games, listening to music, watching TV, making calls, chatting online, browsing social media, watching clips, posting/sharing, using dictionaries, taking online courses, finishing homework, etc.). Detailed information on the items is presented in Table 2. This questionnaire used a 4-point Likert scale (1 = never, 4 = frequently). Each item indicates an independent construct of a specific SUC, and one-item measurement usually has a similar construct to multiple-item measurement [26]. The use scale measures 20 different SUCs.

#### 2.2.3. Well Being

*Depression*. The Center for Epidemiologic Studies Depression Scale (CES-D-10) [27] was used to measure adolescents’ depression. The CES-D-10 scale contains 10 items (e.g., “I felt depressed”) and it applies a 4-point Likert scale (1 = “rarely or none of the time (<1 day)”, 4 = “most or all of the time (5–7 days)”) to rate the items (α = 0.84). Two items with reversed score were reversed when calculating mean score of the scale. The CFA of CES-D-10 showed good structural validity (CFI = 0.976, TLI = 0.966, and RMSEA = 0.073 (90% CI [0.068, 0.079])).

*Anxiety*. The Generalized Anxiety Disorder-7 scale (GAD-7) [28] was used to measure adolescents’ anxiety. The GAD-7 includes 7 items (e.g., “Feeling nervous, anxious or on edge”) and uses a 4-point Likert scale from 1 (“Not at all”) to 4 (“Nearly every day”) to rate the items (α = 0.96), and the CFA showed good structural validity (CFI = 0.993, TLI = 0.988, and RMSEA = 0.066 (90% CI [0.057, 0.076])).

*Somatization*. The Somatic Symptom Scale-8 (SSS-8) [29] was used to measure adolescents’ somatization. The SSS-8 contains 8 items (e.g., “Stomach or bowel problems”). It uses a 5-point Likert scale from 0 (“Not at all”) to 4 (“Very much”). to rate the items (α = 0.95), and the CFA showed good structural validity (CFI = 0.987, TLI = 0.978, and RMSEA = 0.074 (90% CI [0.066, 0.082])). 

The mean score of each scale was used to represent adolescents’ mental health and used for later analysis. The reliability and validity of CES-D-10, GAD-7, and SSS-8 were confirmed among Chinese adolescents and were widely used by Chinese researchers [30,31].

#### 2.2.4. Covariates

The covariates in this study consisted of adolescents’ gender, age, socioeconomic status (both composite and perceived), smartphone use time on weekdays and weekends, annual family revenue, parents’ educational level, and baseline mental health. Of these covariates, parents only responded to their annual family income, educational level (that of both the father and the mother), and the SES of the family (perceived SES). Other covariates were obtained from adolescents. One item was used to measure perceived SES—“How would you rate your family’s socioeconomic status in this city?” (the item uses a 5-point Likert scale (1 = “very bad”, 5 = “very good”)). The composite SES was computed by the average of the standardized scores of educational levels of parents and annual family income [32]. Smartphone use time on weekdays and weekends ranged from no use (1) to 7 h or more (7). Finally, to control the auto-regressive effect of the dependent variables, the baseline mental health (the average value of anxiety, depression, and somatization) was regarded as a covariate.

### 2.3. Analytic Procedure

First, SPSS 20.0 was used to preprocess the data, identify missing data, and calculate the reliability coefficient of each scale. Second, to explore the relationship between SUC and mental health, and to reduce the subjective bias and arbitrariness of researchers, we conducted a 3 step specification curve analysis [18]: (1) Defining a set of reasonable specifications. If the defined specifications are not theoretically or empirically reasonable, it would cause “truly arbitrary” issues, which might impact the results. To define all the specifications, the researcher should calculate all non-redundant combinations (specifications) of different independent, dependent, and covariate variables. In this study, the included variables (e.g., independent variables, dependent variables, and covariates) are chosen based on previous literature or theories. (see the previous paragraph regarding measurements). (2) Estimating all reasonable specifications and describing the outcomes. To estimate all specifications and describe them, a linear model is used to calculate the regression coefficients across the different specifications, then all the coefficients were distributed in an ascending curve. (3) Conducting joint statistical tests. The joint statistics test analyzes whether the actual results were inconsistent with the results under the null hypothesis. Consistent with previous practice via forcing the null on the existing data (the original data), we created the null hypothesis datasets (the null hypothesis was true). The null dataset consists of the original independent variables and covariates but contains newly constructed dependent variables via forcing the null on original data. Thus, there is no relationship between independent variables and dependent variables in the null dataset. Then, participants in the null dataset are randomly drawn with replacement, creating a new SCA model based on the null datasets (the new SCA model is a null hypothesis model in which there exists no relationship between independent variables and dependent variables). After 500 times, the SCA models that tested the effects of SUCs on mental health (which turned out to have no significant effect) are constructed. Several statistical indicators based on the original SCA (computed from the original dataset) were compared with the indicators from the SCA (computed from the null dataset) to find whether significance exists. The used statistical indicators in this study included the median β (median regression coefficient) and the number of significant results in the predominant direction (NSRPD) [18]. Although some researchers have used only one of them to determine the significance of the SCA [33], the prediction effect was robust when both the median regression coefficient and the NSRPD were significant [18]. For more information on SCA, the reader is referred to the tutorial [18]. The R package *specr* was used to estimate and present the specification curve analysis [34]. All the variables were standardized prior to SCA. In longitudinal studies considering autoregressive effects, the effect size is small when β is between 0.03 and 0.07, medium when β is between 0.07 and 0.12, and large when β is greater than 0.12 [35].

## 3. Results

### 3.1. Identification of Specifications and Description of the Estimate

As shown in Figure 1B, the identified specifications total 540 for this curve (identified specifications = SUCs (twenty choices) × mental health (three choices) × covariate (nine choices)). The descriptive specification curve displayed the distribution of the estimates obtained through alternative reasonable specifications and identified which analytical decisions are the most consequential [18]. Figure 1A presents the estimates and describes the results from the longitudinal data. A total of 386 specifications obtained positive and significant regression coefficients, two specifications obtained negative and significant results, and 152 specifications showed no significance. The results show that the regression coefficient varies from −0.05 to 0.20. 

### 3.2. Statistical Inference

The statistical inference based on bootstrapped null models is presented in Table 3. In this longitudinal data, only adolescents’ online courses showed no predictive effect on their mental health (median β = 0.01, *p* > 0.05, NSRPD = 0/27, *p* > 0.05). Finishing homework via smartphones also shows a negligible effect (median β = 0.03, *p* < 0.05, NSRPD = 8/27, *p* > 0.05). Watching TV (median β = 0.14, *p* < 0.001, NSRPD = 24/27, *p* < 0.05), playing games (median β = 0.09, *p* < 0.001, NSRPD = 19/27, *p* < 0.05), listening to music (median β = 0.18, *p* < 0.001, NSRPD = 25/27, *p* < 0.05), and chatting online (median β = 0.15, *p* < 0.001, NSRPD = 24/27, *p* < 0.001) showed negative medium and high prediction for future mental health. The left 15 types of content (e.g., using utilities, using cameras, obtaining life information, making calls, and consuming online) during smartphone use showed significant and negative effects for future mental health only with the criterion of median β (β = 0.05 to 0.14, *p* < 0.05) but showed no significance with the standard of the NSRPD (*p* > 0.05).

The effects of SUC on adolescents’ mental health ranged from none (e.g., participation in online courses and homework), small (e.g., playing games) to medium (e.g., watching TV), depending on the SUCs.

## 4. Discussion

Using SCA, our results show that different types of SUC are differently associated with depression, anxiety, and somatization in adolescents. Depending on the types of content used, these effects ranged from high, medium, and small to none. The relatively descending order of effect on mental health is listening to music, chatting online, watching TV, playing games, types of content (e.g., browsing social media, making payments, and reading online novels) with high but unstable effects, types of content with medium (e.g., browsing news, posting/sharing) but unstable effects, types of content (e.g., using the camera, obtaining life information, and making calls) with small but unstable effects, such as finishing homework and taking online courses.

Our longitudinal results support the view that digital technology use is related to low mental health. However, our results are inconsistent with previous similar research arguing that adolescents’ screen time has no relationship with their well being [36]. Researchers believe that the use of content rather than screen time may be related to mental health [14]. Our result is also inconsistent with another study by Orben and Przybylski arguing that although the effect of technology use is negatively associated with well being, it is too low to be considered [5]. However, the predicted regression coefficients of smartphone content in this study are larger than those of the previous research focusing on screen time. This may be because the current study has focused on smartphones and their content, and since smartphones have become increasingly popular, the negative effect of smartphone use can be significantly elevated [6]. The datasets used in previous research [5] were collected before the wide popularization of smartphones, which may have contributed to the obtained minus effects (extremely low regression coefficients) [6]. For example, a recent study has found that longer screen time has strong negative effects on mental health [37]. Experimental research has also found that limiting social media use can decrease loneliness and depression [38].

We found that entertainment activities (such as playing games, watching TV, and listening to music) are strongly related to adolescents’ mental health, which is consistent with previous research [39] and supports our hypothesis. Researchers have explained that owing to their attractiveness, entertainment activities may contribute to adolescents’ excessive smartphone use [40,41], which may lead to loss of control, disrupt their daily lives, and negatively affect their mental health [42]. Interpersonal activities (e.g., chatting online) on smartphones also positively predicted adolescents’ depression, anxiety, and somatization, which provides more evidence to support studies focusing on social network use [43]. The pervasive connectivity brought by smartphones has advantages in providing adolescents with interpersonal communication channels, which are conducive to their mental health. However, the complementary–interference model also points out that online chatting and entertainment activities might supplant face-to-face interactions and distract face-to-face interaction, which will offset the potential advantages of smartphones on mental health [21].

One interesting finding is that taking online courses through smartphones exerts no effect on adolescents’ mental health while doing homework is slightly and negatively related to mental health. Although the results do not fully support our hypothesis, at least, the negative effect of doing homework via smartphones is negligible. We suggest that adolescents usually complete their homework alone while collaboration and interaction with others during online courses may provide potential social support [22], which may benefit their social adjustment. Furthermore, the left 15 types of smartphone content (e.g., using search engines, using cameras, and posting/sharing) only show significance in median β not in NSRPD, which means that although their effect size is medium or large while such effects may not be very stable. We think this may be related to the intensity of interference when using different smartphone content and related contexts. For example, playing games, watching TVs, and chatting online need plenty of cognitive resources and usually take place in leisure time with real-life others and need a lot of time to complete [20], which might cause more interference with their daily sources of well-being. Although obtaining life information, using fitness apps, making calls, and making payments might interfere with their life, these types of content exert fewer effects on mental health because of less cognitive resources allocation and less time needed. As for using smartphones for learning and homework demanding many cognitive resources, these activities usually take place in an environment with few interactions with real-life others, which is less likely to disturb their sources of well being. However, why the effects of the left 15 types of SUC are unstable, and what reasons may contribute to this, future studies may focus on this. 

This study has some potential practical implications. Different types of SUC show different effects on adolescents’ mental health, which can inform the practice of policymakers and parental media mediation. Policy statements regarding screen-based media exposure for children and adolescents should carry a distinction between different SUC, rather than proposing a blanket limit on screen time (as recommended by the American Academy of Pediatrics). When it comes to adolescents’ smartphone engagement, policymakers or parents might propose or use different mediation strategies towards different functions or Apps on smartphones. For example, policymakers can encourage loose screen time on mental health-friendly smartphone functions (with a negligible and stable effect size), such as taking online courses and finishing homework; at the same time, advocate restricted screen time on mental health-harmful smartphone functions (with a high and stable effect size), such as watching TV, playing games, listening to music, and chatting online. As nutritionists often do, a “nutritionally balanced” digital diet for young people should be recommended to the public, rather than just suggesting limits on the amount of time they can spend using digital media.

This study has some limitations. First, all the measurements are subjective evaluations and may produce some bias. Second, as Twenge argued that the definition of mental health or well being should be defined broadly, it may include positive emotions (happiness and life satisfaction), and indicators of negative emotions (depression and suicide attempts) [6]. In this study, we only measured negative emotions, and, thus, future studies should include both positive and negative indicators and use SCA to provide more evidence. Third, the relationship between adolescents’ mental health and smartphone content may be bidirectional. The present study failed to answer such an issue because our scope focused on the effect of SUC on adolescents’ mental health. Fourth, although we stress the smartphone content, we fail to provide an answer as to why some content is associated with negative mental health, such as using search engines. Future studies should focus on what is searched by adolescents via smartphones which may provide some explanations, and more qualitative studies are required in the future. Finally, the effect size for some SUCs is large, but such effects might not be very stable, more confirmation is needed in the future (e.g., using a nationally representative sample size). 

## 5. Conclusions

Some SUCs are negatively and longitudinally related to adolescents’ mental health. Smartphones used for listening to music, chatting online, watching TV, and playing games are negatively and longitudinally related to mental health in adolescents. Using smartphones for online courses is uncorrelated with adolescents’ mental health. The left 15 SUCs (e.g., using search engines or reading online novels) reveal unstable effects. Researchers and policymakers should update their understanding of adolescents’ technology use, especially to adopt a differentiated attitude or different policies towards different media use content and a more cautious attitude towards the overall screen-time restriction policy.

## Figures and Tables

**Figure 1 children-10-00770-f001:**
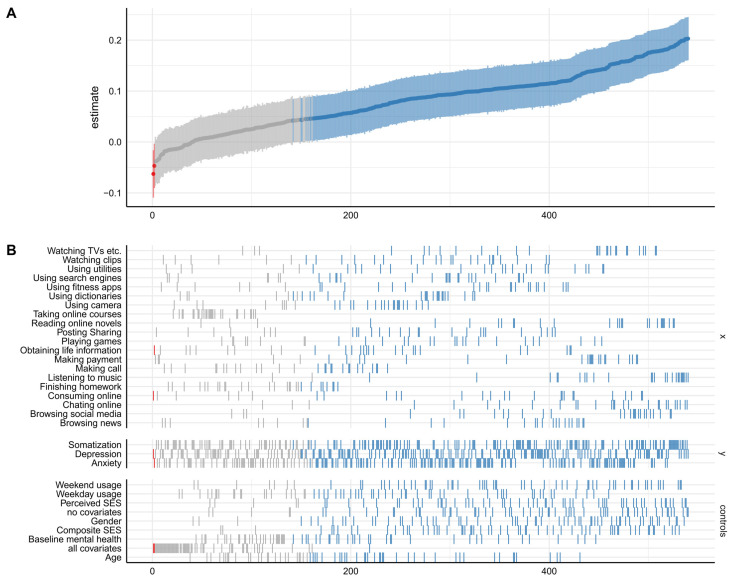
The description of the specification-curve analysis. *Note*. The estimate represents the standardized regression coefficients, the *x*-axis represents the number of specifications, and the *y*-axis in (**B**) represents the independent, dependent, and control variables. Red points represent negative and significant coefficients (significance level α = 0.05). Blue points represent positive and significant coefficients. Grey points refer to nonsignificant coefficients. The curve in (**A**) is the distribution of coefficients, and the shaded parts around the curve suggest 95% confidence interval.

**Table 1 children-10-00770-t001:** Demographic information of adolescents.

Variables	Groups	Percentage (%)
Residence	City	47%
	Township	15.3%
	Rural region	37.7%
Only child	Yes	91%
	No	9%
Mother’s education	<College	90.5%
	≥College	9.5%
Father’s education	<College	87.5%
	≥College	11.4%
Annual income	<50,000 ¥ ^1^	60.2%
	50,000–100,000 ¥	21.6%
	>100,000 ¥	19.2%

^1^ ¥ = RMB.

**Table 2 children-10-00770-t002:** The abbreviation of SUC items.

Abbreviation of SUC Items	M ± SD
1. Watching TV, etc.	2.12 ± 0.82
2. Watching clips	2.07 ± 0.85
3. Using utilities	2.21 ± 0.80
4. Using search engines	2.16 ± 0.76
5. Using fitness apps	1.77 ± 0.78
6. Using dictionaries	2.16 ± 0.76
7. Using camera	2.05 ± 0.70
8. Taking online courses	2.28 ± 0.75
9. Reading online novels	1.88 ± 0.82
10. Posting/sharing	1.64 ± 0.71
11. Playing games	2.01 ± 0.83
12. Obtaining life information	2.17 ± 0.83
13. Making payments	1.91 ± 0.91
14. Making calls	2.16 ± 0.73
15. Listening to music	2.35 ± 0.89
16. Finishing Homework	2.06 ± 0.76
17. Consuming online	1.58 ± 0.72
18. Chatting online	2.30 ± 0.88
19. Browsing social media	2.00 ± 0.83
20. Browsing news	1.99 ± 0.91

Note: JD and Taobao = online shopping apps. WeChat and QQ = online chat apps. Qzone, Weibo, and WeChat Moments = social media apps. Kuaishou, Volcano video, and TikTok = video-sharing social networking apps. The full information on SUC items can be found in the work of Huang et al. [25], permission has been obtained and there is no copyright issue. For example, the full item description for Taking online courses is “Use a mobile phone to learn online courses” and Posting/sharing is “Post or share pictures, videos, audio, songs, articles, or applets created by yourself”.

**Table 3 children-10-00770-t003:** Results of the SCA bootstrapping tests.

Independent Variables	Median Point Estimate (β)	NSRPD
Watching TV, etc.	0.14 ***	24/27 *
Watching clips	0.09 ***	21/27
Using utilities	0.10 ***	22/27
Using search engines	0.09 ***	20/27
Using fitness apps	0.10 ***	21/27
Using dictionaries	0.09 ***	21/27
Using cameras	0.07 ***	19/27
Taking online courses	0.01	0/27
Reading online novels	0.15 ***	24/27
Posting/sharing	0.08 ***	20/27
Playing games	0.09 ***	19/27 *
Obtaining life information	0.06 ***	18/27
Making payments	0.14 ***	22/27
Making calls	0.05 ***	14/27
Listening to music	0.18 ***	25/27 *
Finishing homework	0.03 *	8/27
Consuming online	0.10 ***	21/27
Chatting online	0.15 ***	24/27 *
Browsing social media	0.16 ***	24/27
Browsing news	0.11 ***	19/27

Note. * *p* < 0.05, *** *p* < 0.001.

## Data Availability

The data presented in this study are available on request from the corresponding author. The data are not publicly available due to ethical restrictions.

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
