# Peer review of "Beyond Screen Time: The Different Longitudinal Relations between Adolescents’ Smartphone Use Content and Their Mental Health"

_children, 2023, doi:10.3390/children10050770_

Round 1

Reviewer 1 Report

Dear Authors,

Firstly, congratulation on your very interesting research Beyond screen time: The different longitudinal relations between adolescents’ smartphone use content and their mental health

Secondly, I am the opinion, that some parts of the manuscript need a minor revisions.

Introduction: please make it more concise. In this form it is too long.

M&M: lines 110- 130: this part should be divided into: study design, recrutiment to the study and ethical issues. What is the no. of the Institutional Review Board consent? 

Lines 156-169. Well-being: the authors should elaborate more about Epidemiologic Studies Depression Scale (CES-D-10), the Generalized Anxiety Disorder-7 scale (GAD-7) and the Somatic Symptom Scale-8 (SSS-8), being each in three separate paragraphes.

Discussion: Could the authors elaborate more about some practical implications that could be drawn from results of the study?

Author Response

Best wishes

Reviewer 2 Report

Dear authors,

I think the manuscript is well-written, understandable, and is coherent with what the literature says about the relationship between smartphone use and psychological wellbeing. However, I think the article does not provide useful results from a practical point of view.

It is true that smartphone way of use will depend on the content, and it will influence their wellbeing. Besides, it is known that it is a complex and multifactorial phenomenon. Perhaps you provide a new approach to it using a sophisticated data analysis. However, the conclusions you reach do not provide new results that will be useful from a practical point of view. Your contributions are theoretical (already existing) but do not provide practical contributions. How these results can be used for practical issues?

In your manuscript title and in the research description you point out that the research has a longitudinal approach. Nevertheless, I miss a comparison between the two measure times. What information do you want to provide with these two measures?

Specification curve analysis is a very new data analysis and researchers and readers are not familiarized with it. In this sense, your explanation about it is lax. For example, on Analytic procedure section you point out (line 190) “Defining a set of reasonable specifications”. Then, you provide some more information, but I think it is not enough. What are reasonable specifications and why? On the Introduction section you mention that (line 84) “… aims to explore the robust and different effects…”. Which is the output of that analysis that inform about the robustness of the results? Besides, information provided on Table 2 it is not comprehensive.  I think it is needed a deep explanation of that.

There are variables that have been measured, but you do not say anything about that. For example, these is the case of sociodemographic variables.

I have other comments about your manuscript:

Lines 63-65: I do not think that the reproducibility crisis is responsible of the controversial results. In any case, I think you should justify your statements about it. 

Line 66: For sure, researchers choose those variables they are interested in. I do not think that this statement “… (SCA) can be used to alleviate this problem…” will solve the problem, because choosing variables is an arbitrary phenomenon. It is impossible to consider all the variables involved in the smartphone use, due its multifactorial character.

There are a too long sentence between lines 90 to 96. Perhaps you can divide it for an easy understanding.

There is a missing “s” at the end of “Participant” section title.

I think you should indicate the number of participants surveyed in the first and the second year, to know if there are missing ones.

Which are the number of fathers and mothers participating in the research? Which is the purpose of considering them to the research?

You do not provide information about reliability of the Mobile Phone Use Pattern Scale for Chinese students instrument used in your research.

When you describe well-being measures, I miss a brief explanation about how to obtain the total score in all cases, and if there are reverse items to take into account.

There are minor details as missing spaces between text and “[“ symbol when citing, or using italics for “p-values” in some parts of the text but not in others.

At last, but not least, on the Conclusions section you write the information in a future way, but you have been writing in the present in all the text. I do not understand that. Have your data the capability of future prediction? Or, in other words, do the SCA provide information about causality and allow to predict future results? I think you must circumscribe your results in the time and participants in which you collect the data, although it will be useful to understand the phenomenon you are investigating.

That is all from me. I hope you find my comments useful to improve your manuscript.

Best wishes,

Dr-05

Author Response

Best wishes

Reviewer 3 Report

This is an interesting study examining the different longitudinal relations between adolescents' smartphone use content and their mental health. The study is well-conducted and I agree that it may contribute well to the literature. I only have a few comments to improve the manuscript further:

1. In the second paragraph of the introduction, I would suggest the authors to also argue that many of the existing studies in the relationship between smartphone/social media and mental health were cross-sectional which are vulnerable to reverse causation issue. This will highlight the need for more well-conducted longitudinal study that is actually a big strength of the current study.

Please refer to this relevant review: Does social media use increase depressive symptoms? A reverse causation perspective. (2021). Frontiers in Psychiatry, 12, 641934.

2. The inclusion and exclusion criteria of the participants should be made clearer in the method section. The authors should also report on how the sample was recruited in the current study.

3. The descriptive statistics for smartphone related measures (e.g., types of smartphone use content) should be reported in the method section too.

4. In the analytic procedure, I appreciate the fact that the authors highlighted how effect size can be categorized. However, there is a lack of discussion related to effect sizes found in the current study in the Discussion section.

Author Response

Best wishes

Round 2

Reviewer 2 Report

Dear authors,

I think that the methodological part could be further clarified. In any case, the manuscript has improved with the changes implemented.

Best,
Dr-05

Author Response

Dear reviewer,

Thank your previous and present comments to our manuscript to improve it. 

As for the method part, we revised this part. The unclear sentences or logical issues were checked, and corresponding revision were made. Revision regarding this part were highlighted.

Besides, English grammar and spelling were checked again, all the revision were highlighted.

Thank you,

Best wishes.

Reviewer 3 Report

The authors have addressed all my comments well. I appreciate all their efforts.

Author Response

Dear reviewer,

Thank your previous and present comments to our manuscript to improve it. 

As for the mention in clearly presenting our results, we checked our results and table 3 again. And some potential misunderstandings were corrected. Corrections regarding the results part were highlighted.

Besides, English grammar and spelling were checked again, all the revision were highlighted.

Thank you,

Best wishes.